# Trajectory of the arterial-alveolar oxygen gradient in COPD for a decade

Kazuma Nagata[1], Susumu Sato [1,2]\*, Kiyoshi Uemasu[1©], Naoya Tanabe[1], Atsuyasu Sato[1], Shigeo Muro[3©], Toyohiro Hirai [1]

1 Department of Respiratory Medicine, Graduate School of Medicine, Kyoto University, Kyoto, Kyoto, Japan, 2 Department of Respiratory Care and Sleep Control Medicine, Graduate School of Medicine, Kyoto University, Kyoto, Kyoto, Japan, 3 Department of Respiratory Medicine, Nara Medical University, Kashihara, Nara, Japan

© These authors contributed equally to this work.
\* ssato@kuhp.kyoto-u.ac.jp

## Abstract

### Background

Chronic respiratory failure (CRF) is a critical complication in patients with chronic obstructive pulmonary disease (COPD) and is characterized by an increase in the arterial-alveolar oxygen gradient ($A\text{-}aDO_2$). The long-term trajectory and prognostic significance remain unclear. This study aimed to assess the prognostic impact of $A\text{-}aDO_2$ and elucidate its trajectory over ten years.

### Methods

We enrolled 170 outpatients with COPD from a prospective cohort study. Arterial blood gas (ABG) analyses were conducted annually for ten years while monitoring the development of CRF.

### Results

157 patients completed the observation period, of whom 21 developed CRF (CRF group) and 136 did not (non-CRF group). In the CRF group, there was a gradual increase in $A\text{-}aDO_2$ along with decreases in partial pressure of oxygen ($PaO_2$) and partial pressure of carbon dioxide ($PaCO_2$) over ten years, although there were no changes in the non-CRF group. The CRF group had higher baseline $A\text{-}aDO_2$ and higher $\Delta A\text{-}aDO_2$ in the first year than the non-CRF group (3.76 vs. 0.42 Torr/year, p = 0.030). Kaplan-Meier analyses, and multivariate Cox proportional hazards analysis revealed that both baseline $A\text{-}aDO_2$ and $\Delta A\text{-}aDO_2$ were significantly associated with the development of CRF. Retrospective tracking from the initiation of long-term oxygen therapy (LTOT) revealed significant increases in $A\text{-}aDO_2$ from 5 years prior to LTOT initiation in the CRF group when compared to the non-CRF group.

**Data Availability Statement:** "Data cannot be shared publicly because of patient disagreement. Data sets are available upon request, and we need to get approval again from the ethical committee of our institute (Kyoto University Graduate School and

Faculty of Medicine, Ethics Committee) to share data including patient data (even de-identified or anonymized data). The contact email address for the ethics committee is "ethcom@kuhp.kyoto-u.ac.jp". Inquiries regarding Data Sharing should be directed to the Clinical Research Consultation Office at Kyoto University Hospital, contacting address is "ctsodan@kuhp.kyoto-u.ac.jp"."

**Funding:** This work was supported by JSPS KAKENHI Grant Numbers 19K08624, 16K09536, 25461156, 21590964, 16390234, JP22K11391 and JP22H04411. This research was also supported by grants from the Intractable Respiratory Diseases and Pulmonary Hypertension Research Group of the Ministry of Health, Labour and Welfare of Japan (JPMH20FC1027, JPMH23FC1031). The funders had no role in study design, data collection and analysis, publication decisions, or manuscript preparation.

**Competing interests:** S.S. reports a grant from Nippon Boehringer Ingelheim Co. and grants from Philips-Respironics, Fukuda Denshi, Fukuda Lifetec Keiji, and ResMed that did not pertain to the submitted work. K.N. reports a honoraria for lectures at Teijin Healthcare unrelated to the submitted work. S.M. reports a grant from ROHTO Pharmaceutical unrelated to submitted work. This does not alter our adherence to PLOS ONE policies on sharing data and materials.

## Conclusions

An increasing trend in A-aDO$_2$ may be a significant sign for the future development of CRF. A transition of the annual change of A-aDO$_2$ from a stable state to a deterioration phase can serve as a prognostic factor for developing CRF within 5 years.

## Introduction

Chronic obstructive pulmonary disease (COPD) is a destructive lung disease characterized by chronic respiratory symptoms due to abnormalities of the airways and/or alveoli that cause persistent, often progressive, airflow obstruction [1]. Advanced COPD is often accompanied by chronic respiratory failure (CRF) and is characterized by hypoxemia.

CRF in COPD patients is a critical complication associated with reduced quality of life, decreased tolerance for exercise, and increased risk of death [2, 3]. Arterial blood gas (ABG) analysis is a crucial diagnostic tool for managing CRF associated with respiratory diseases, including COPD. Our previous prospective cohort study involving patients with COPD examined the importance of ABG parameters in predicting six-year CRF development [4]. We found no progression to CRF in the group with normal partial pressure of oxygen (PaO$_2$) levels (PaO$_2 \geq 80$ Torr) and focused our investigation solely on the group with low PaO$_2$ levels (PaO$_2 < 80$ Torr). The results indicated that the annual change in PaO$_2$ is able to predict CRF development over six years. However, due to the short duration of the study, a longer follow-up is required to reach definitive conclusions for the entire cohort, including those with normal PaO$_2$ levels. We also consider the alveolar-arterial oxygen gradient (A-aDO$_2$) a promising parameter.

A-aDO$_2$, a parameter derived from ABG analysis, is an indicator of the efficiency of gas exchange in the lungs; it is defined as the difference between the partial pressure of oxygen in the alveoli (P$_A$O$_2$) and PaO$_2$. The elevated A-aDO$_2$ associated with hypoxemia is a good indicator of ventilation/perfusion mismatch and intrapulmonary shunting [5]. Previous studies have applied A-aDO$_2$ for patients with several acute respiratory diseases, such as community-acquired pneumonia and coronavirus disease 2019 (COVID-19), and as an indicator of disease severity and outcome [6–9]. Nonetheless, the prognostic implications of A-aDO$_2$ in patients with COPD have been scarcely explored, and the longitudinal natural course of A-aDO$_2$ in relation to the progression of COPD has also been inadequately examined.

In this study, our primary objective was to extend our previous cohort study to a ten-year follow-up and determine whether the annual changes in ABG parameters, especially A-aDO$_2$, could more accurately predict the future onset of CRF across the entire cohort. For this purpose, we compared the physiological parameters and their yearly changes between patients who developed CRF over the decade and those who did not. Additionally, considering that some COPD patients develop CRF over the long term while others do not, we hypothesized the potential existence of a 'transition' from stable to exacerbation phases in the disease course and sought to validate this hypothesis. In addressing this, we aimed to compare the continuous longitudinal changes in ABG parameters over ten years between each group, focusing on assessing their prognostic significance.

## Materials and methods

### Study subjects

We collected data for stable COPD patients who visited Kyoto University Hospital between January 2006 and November 2008 and consented to ABG testing [4]. These patients were in

exacerbation-free periods, with no exacerbations reported in the four weeks prior. Each patient had a smoking history of $\geq 20$ pack-years and had been diagnosed with COPD according to the Global Initiative for Chronic Obstructive Lung Disease (GOLD) criteria [1]. We collected information about COPD exacerbations in the preceding year by reviewing medical records and patient self-reports. COPD exacerbation was defined as increased COPD symptoms requiring treatment with antibiotics or additional systemic corticosteroids. However, certain patients were excluded from the study: (1) those utilizing long-term oxygen therapy (LTOT); (2) those with chronic heart or renal failure; (3) those with coexisting respiratory disorders, such as bronchial asthma, interstitial pneumonia, or chronic respiratory infection; and (4) those with a history of thoracic surgery or cancer therapy. Additionally, patients with a baseline $PaO_2 < 60$ Torr were excluded.

The data utilized in this study were derived from a prospective cohort approved by the ethics committee of Kyoto University (R0311-4). Prior to the commencement of the study, all participants in the cohort provided either written or oral informed consent. This trial was conducted according to the principles of the Declaration of Helsinki and the Ethical Guidelines for Medical and Health Research Involving Human Subjects (Japanese Ministry of Health, Labor and Welfare). Data including medical records of all participants were accessed from Sep 1st 2023 to Dec 15th 2023.

## Measurements

The patients underwent baseline assessments at the beginning of the study. These assessments included pulmonary function tests, ABG analysis, and evaluations of dyspnoea symptoms. Based on these assessments, we designed a comprehensive treatment plan encompassing both pharmacological treatments and nonpharmacological interventions, such as smoking cessation. In addition, patients received verbal guidance on potential methods to maintain or enhance physical activity levels.

Following their enrolment, the patients regularly visited our outpatient clinic every 1 to 3 months. During these visits, continuous pulse oximetry and symptom evaluations were performed to monitor disease status and identify the need for LTOT. The need for LTOT was determined by the attending physician, primarily based on the presence of progressive resting hypoxemia ($PaO_2 \leq 55$ Torr or percutaneous oxygen saturation ($SpO_2$) $\leq 88\%$). ABG analyses were scheduled annually, with the first follow-up conducted one year after baseline. Subsequent assessments were aligned with stable phases, defined as at least four weeks after recovery from any exacerbation, to minimize variability due to acute changes. One year after the initial assessment, we performed additional pulmonary function tests and ABG analyses to monitor the progression of ABG parameters and other pulmonary function tests. Subsequently, we carried out annual ABG analyses for the next nine years for a total monitoring period of ten years. As ABG analyses were undertaken while using oxygen after initiation of LTOT, ABG measurements obtained post-LTOT initiation were excluded from our assessment.

For all ABG analyses, we collected samples after the patient had rested for 15 minutes. We used a RAPIDlab 1265 Blood Gas Analyzer (Siemens Healthcare Diagnostics Inc., Malvern, PA, USA) to measure arterial pH, $PaO_2$, and the partial pressure of carbon dioxide ($PaCO_2$). Pulmonary function tests, which included spirometry, lung volume subdivisions and diffusion capacity assessment ($DL_{CO}$), were conducted using a Chestac-65V instrument (Chest MI, Tokyo, Japan). Predicted pulmonary function values were calculated in accordance with guidelines provided by the Japanese Respiratory Society [10].

During the ten-year observation period, we closely tracked the patients in our study. The primary outcome was the development of CRF, as indicated by the initiation of LTOT;

mortality was the secondary outcome. Patients who began LTOT were diagnosed with CRF, with accompanying hypoxemia, which served as a crucial outcome measure. In addition, we recorded the date of LTOT initiation for further analysis. In cases in which a patient was hospitalized due to exacerbation, oxygen therapy was initiated, and LTOT was continued post-discharge. The initiation point was defined as the time when oxygen therapy was commenced during the exacerbation.

## Statistical analysis

Continuous variables are expressed as the mean±SD unless otherwise specified. All statistical analyses were performed using JMP 16 software (SAS Institute Inc., Cary, NC, USA). The change in pulmonary function and arterial blood gas parameters over a year was calculated and expressed as ΔParameter (/year). We compared patient characteristics, parameters, and changes in parameters between two groups by Student's t, Fisher's exact, and chi-square tests. Overall survival (OS) was assessed using the Kaplan–Meier survival curves, and differences in OS between the groups were evaluated using the log-rank test. Multivariate Cox proportional hazards analysis was conducted to elucidate the relationships between clinical parameters and the development of CRF. $P$ values less than 0.05 were considered statistically significant. We conducted a post-hoc sample size analysis using data from the present study. For the $\Delta A\text{-}aDO_2$ comparison between the CRF group and the non-CRF group, we calculated the required sample size for $\alpha = 0.05$ and $\beta = 0.2$ to determine if the sample size in our study was adequate to detect significant differences.

## Results

### Patient selection and demographics

Fig 1 shows the study flow chart. We initially registered 220 patients. Then, 170 patients who underwent baseline and 157 who underwent follow-up assessments were selected for final analysis (Fig 1).

We initially compared the baseline characteristics of two groups: the CRF group (n = 21), which needed LTOT within 10 years, and the non-CRF group (n = 136), which did not require LTOT in this same time span. Our findings revealed that compared to the non-CRF group, the CRF group had significantly higher MRC dyspnoea scale scores, lower levels of $PaO_2$, increased $A\text{-}aDO_2$, reduced $FEV_1$, and lower $DL_{CO}$ and $K_{CO}$ values (Table 1).

Next, we compared changes in pulmonary function tests and ABG analyses over one year (Table 2). The results showed that compared to the non-CRF group, the CRF group had higher $\Delta A\text{-}aDO_2$ and lower $\Delta K_{CO}$. In particular, our focus on $\Delta A\text{-}aDO_2$, the primary parameter of interest in this study, showed that it was higher in the CRF group than in the non-CRF group during the first year (3.76±5.24 vs. 0.42±6.65 Torr/year, p = 0.030).

### Long-term trajectories of ABG parameters in CRF and non-CRF groups

Fig 2 demonstrated the 10-year trends of the CRF and non-CRF groups. In the CRF group, there was a gradual increase in $A\text{-}aDO_2$ and decreases in both $PaO_2$ and $PaCO_2$ over the course of ten years. However, no apparent changes were observed in the non-CRF group over the same period.

We also examined the change in $A\text{-}aDO_2$ from baseline in each year up to 10 years observation period (Table 3). While variations were observed, we identified a trend in which the difference in the magnitude of $A\text{-}aDO_2$ changes between the two groups increased as time progressed.

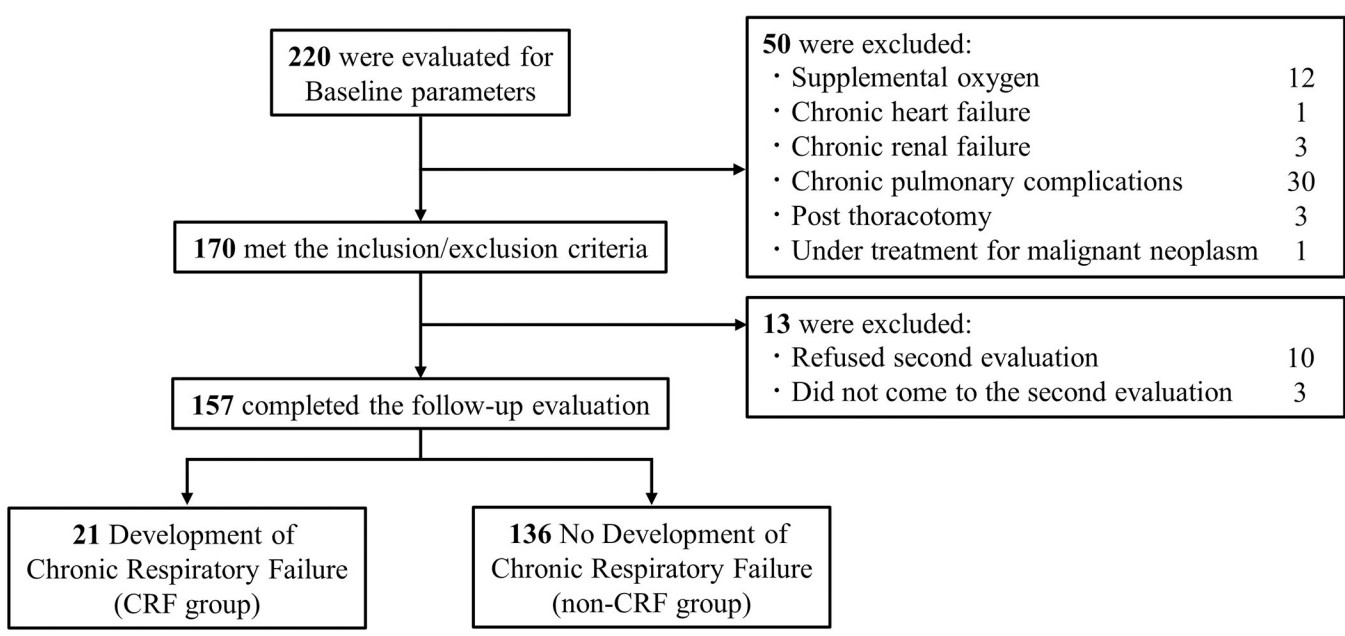

**Fig 1. Study flow chart describing the study enrolment process.** CRF, chronic respiratory failure.

**Table 1. Comparison of background characteristics between patients who did (CRF group) and did not develop CRF (non-CRF group).**

| | CRF group (n = 21) | Non-CRF group (n = 136) | *P* value |
|---|---|---|---|
| Age, years | 73.00±6.12 | 71.04±8.61 | 0.317 |
| Female, n(%) | 1 (4.8) | 14 (10.3) | 0.695 |
| Current smoker, n(%) | 2 (9.52) | 31 (22.79) | 0.250 |
| Smoking index (pack-years) | 72.71±37.73 | 66.69±36.60 | 0.486 |
| BMI (kg/m$^2$) | 20.51±2.51 | 21.68±2.86 | 0.080 |
| mMRC dyspnoea scale (0/1/2/3/4) | 0/13/7/1/0 | 34/70/26/6/0 | 0.009 |
| GOLD (I/II/III/IV) | 3/8/9/1 | 27/70/36/3 | 0.404 |
| PaO$_2$ (Torr) | 71.30±6.33 | 77.63±8.21 | <0.001 |
| PaCO$_2$ (Torr) | 38.47±3.56 | 39.49±4.35 | 0.306 |
| A-aDO$_2$ (Torr) | 30.34±6.75 | 22.73±9.69 | <0.001 |
| FEV$_1$ (L) | 1.33±0.45 | 1.67±0.64 | 0.019 |
| FEV$_1$ (% predicted) | 51.35±15.62 | 62.74±18.13 | 0.007 |
| FVC (% predicted) | 96.99±14.67 | 100.50±18.35 | 0.405 |
| VC (% predicted) | 94.73±13.54 | 93.31±16.91 | 0.714 |
| RV/TLC (%) | 44.29±7.64 | 42.59±7.62 | 0.342 |
| FRC (% predicted) | 98.53±13.86 | 100.69±22.99 | 0.676 |
| DL$_{CO}$ (% predicted) | 38.34±13.84 | 54.20±16.71 | <0.0001 |
| K$_{CO}$ (mL/min/L/mmHg) | 1.98±0.80 | 2.91±0.98 | <0.0001 |
| Prior exacerbations ≥2 | 2 (9.5) | 6 (4.41) | 0.291 |
| Treatment with bronchodilator (LAMA or LABA) | 11 (52.4) | 49 (36.0) | 0.158 |

A-aDO$_2$, alveolar-arterial oxygen gradient; BMI, body mass index; CRF, chronic respiratory failure; DL$_{CO}$, diffusing capacity of carbon monoxide; FEV$_1$, forced expiratory volume in 1 s; FRC, functional residual capacity; FVC, forced vital capacity; GOLD, Global Initiative for Chronic Obstructive Lung Disease; K$_{CO}$, carbon monoxide transfer coefficient; LABA, long-acting β2-agonist; LAMA, long-acting muscarinic antagonist; mMRC, modified British Medical Research Council; PaCO$_2$, partial pressure of carbon dioxide; PaO$_2$, partial arterial pressure of oxygen; RV, residual volume; TLC, total lung capacity; VC, vital capacity.

**Table 2. Comparison of longitudinal changes in arterial blood gas analyses and pulmonary function tests between patients who did (CRF group) and did not develop CRF (non-CRF group).**

|  | CRF group (n = 21) | Non-CRF group (n = 136) | *P* value |
|---|---|---|---|
| $\Delta PaO_2$ (Torr/year) | -1.64±6.37 | 0.63±7.50 | 0.189 |
| $\Delta PaCO_2$ (Torr/year) | -1.69±3.77 | -0.84±4.07 | 0.368 |
| $\Delta A\text{-}aDO_2$ (Torr/year) | 3.76±5.24 | 0.42±6.65 | 0.030 |
| $\Delta FEV_1$ (% predicted) | -1.01±3.43 | -1.47±5.39 | 0.706 |
| $\Delta FVC$ (% predicted) | -2.88±8.22 | -3.72±9.31 | 0.695 |
| $\Delta VC$ (% predicted) | 0.71±8.41 | 2.13±6.88 | 0.393 |
| $\Delta RV/TLC$ (%) | -0.36±5.76 | -1.54±5.25 | 0.345 |
| $\Delta FRC$ (% predicted) | 1.47±9.79 | -3.45±12.71 | 0.091 |
| $\Delta DL_{CO}$ (% predicted) | -2.37±7.73 | 0.78±8.52 | 0.114 |
| $\Delta K_{CO}$ (mL/min/L/mmHg) | -0.15±0.38 | 0.04±0.40 | 0.038 |

A-aDO$_2$, alveolar-arterial oxygen gradient; CRF, chronic respiratory failure; DL$_{CO}$, diffusing capacity of carbon monoxide; FEV$_1$, forced expiratory volume in 1 s; FRC, functional residual capacity; FVC, forced vital capacity; K$_{CO}$, carbon monoxide transfer coefficient; PaCO$_2$, partial pressure of carbon dioxide; PaO$_2$, partial arterial pressure of oxygen; RV, residual volume; TLC, total lung capacity; VC, vital capacity.

## Multivariate Cox proportional hazards analysis of CRF development

We subsequently conducted a Cox proportional hazards analysis to elucidate the relationships between the aforementioned clinical parameters and the development of CRF (Table 4). In the multivariate model, which included factors believed to be predictors of CRF development in COPD patients (e.g., $K_{CO}$, $FEV_1$, age, body mass index (BMI), and smoking index), both baseline A-aDO$_2$ and $\Delta A$-aDO$_2$ (change over the first year) were found to be significantly associated with CRF development. Kaplan-Meier curves for overall survival without CRF among four groups, divided using a two-by-two matrix based on the cutoff values of baseline A-aDO$_2$ (23.75 Torr) and its 1-year change ($\Delta A$-aDO$_2$, 0.86 Torr/year), determined through multivariate ROC curve analysis for optimal sensitivity and specificity, demonstrated significant associations of both baseline A-aDO$_2$ and $\Delta A$-aDO$_2$ with the development of CRF. Additionally, the hazard ratio for the "high&increased" group (high baseline A-aDO$_2$ with high $\Delta A$-aDO$_2$) compared to the "high&sustained" group (high baseline A-aDO$_2$ with low $\Delta A$-aDO$_2$) was 3.03 [95% confidence interval: 1.07 to 8.63] (p = 0.041) (S1 Fig in S1 File).

## Retrospective tracking of ABG parameters from initiation of LTOT

Finally, we retrospectively analyzed the trajectories of ABG parameters, from the initiation of LTOT as the starting point, in the CRF group (Fig 3). We compared this group with 36 patients from the non-CRF group who had completed a 10-year ABG follow-up. In the CRF group, there were no significant differences in PaO$_2$, PaCO$_2$, and A-aDO$_2$ from 10 years to 5 years prior to the initiation of LTOT compared to the non-CRF group; indeed, these values remained stable. However, starting at 5 years before LTOT initiation, a decrease in PaO$_2$ and PaCO$_2$ accompanied by an increase in A-aDO$_2$ was observed in the CRF group, with a significant difference from the non-CRF group. This trend was most prominent regarding A-aDO$_2$. Patient proportions exceeding specific A-aDO$_2$ thresholds each year are detailed in S1 Table in S1 File.

## Post-hoc sample size estimation

The post-hoc sample size analysis indicated that approximately 51 subjects per group (a total of 101 subjects) would be required to achieve sufficient statistical power.

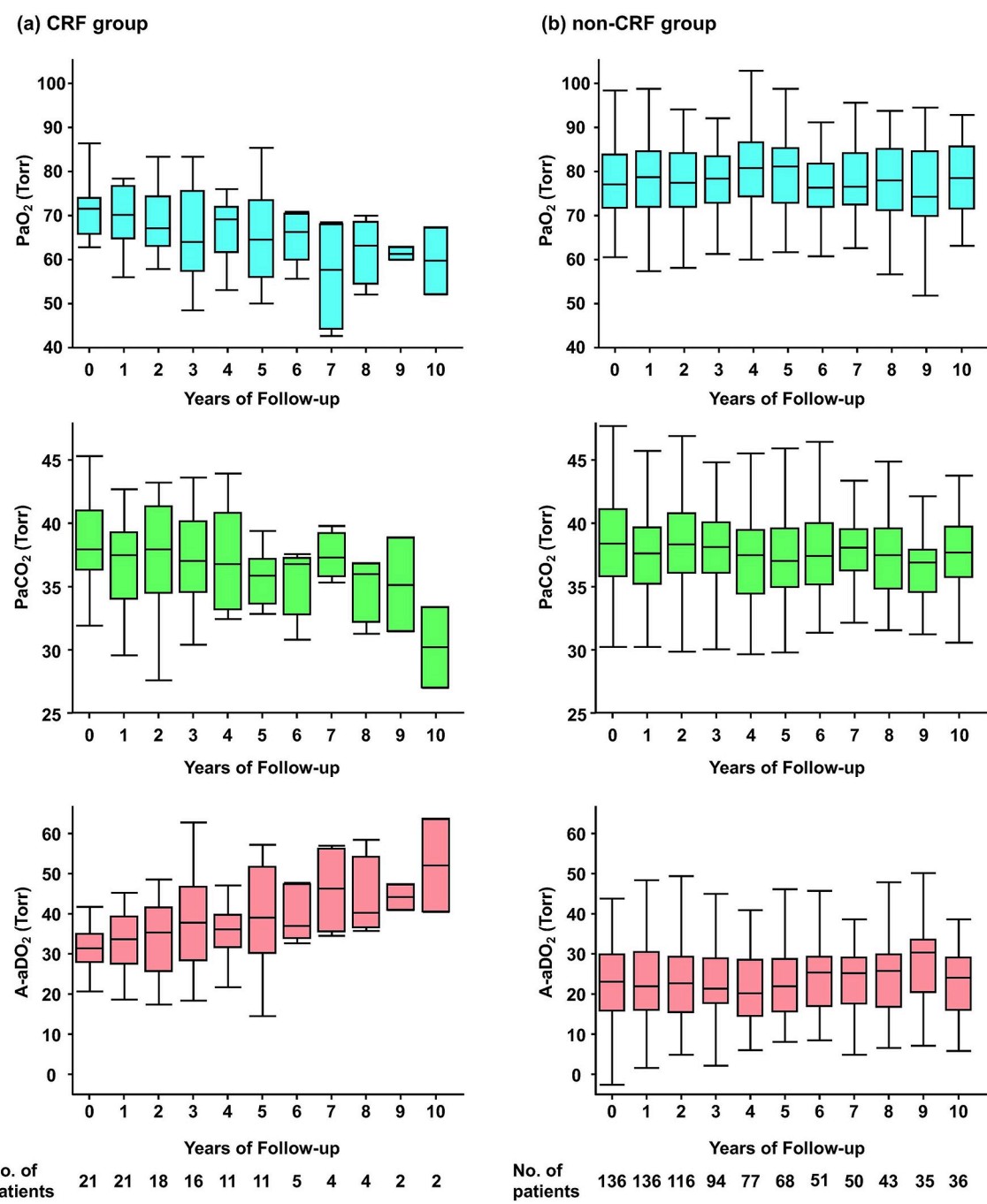

**Fig 2. Long-term trajectories of ABG parameters. (a)** $PaO_2$, $PaCO_2$, and $A\text{-}aDO_2$ in the CRF group. There was a gradual increase in $A\text{-}aDO_2$ and decreases in both $PaO_2$ and $PaCO_2$ over the course of ten years. **(b)** $PaO_2$, $PaCO_2$, and $A\text{-}aDO_2$ in the non-CRF group. There were no apparent changes in any parameter over ten years. $A\text{-}aDO_2$, alveolar-arterial oxygen gradient; ABG, arterial blood gas; CRF, chronic respiratory failure; $PaCO_2$, partial pressure of carbon dioxide; $PaO_2$, partial arterial pressure of oxygen.

## Discussion

In this study, we investigated the clinical importance of ABG parameters, including $A\text{-}aDO_2$, from a longitudinal observation cohort study in patients with COPD and found that the change in $A\text{-}aDO_2$ during the first year might be a potential predictor for progression to CRF

**Table 3. Change in A-aDO$_2$ from baseline in each year among the CRF group and the non-CRF group.**

|  | CRF group | Non-CRF group | *P* value |
|---|---|---|---|
| 1 year (Torr) (n = 21, 136) | 3.76±5.24 | 0.42±6.65 | 0.030 |
| 2 years (Torr) (n = 18, 116) | 3.71±7.05 | 0.10±7.66 | 0.063 |
| 3 years (Torr) (n = 16, 94) | 8.12±10.65 | 0.31±7.93 | <0.001 |
| 4 years (Torr) (n = 11, 77) | 5.77±8.55 | -0.36±7.60 | 0.016 |
| 5 years (Torr) (n = 11, 68) | 10.49±9.76 | 0.17±7.41 | <0.001 |
| 6 years (Torr) (n = 5, 51) | 11.03±7.09 | 1.14±7.14 | <0.005 |
| 7 years (Torr) (n = 4, 50) | 15.21±12.01 | 2.31±7.91 | <0.005 |
| 8 years (Torr) (n = 4, 43) | 14.99±15.36 | 1.25±6.63 | <0.001 |
| 9 years (Torr) (n = 2, 35) | 12.61±3.84 | 3.53±7.26 | 0.091 |
| 10 years (Torr) (n = 2, 36) | 26.17±8.77 | -0.47±6.14 | <0.0001 |

A-aDO$_2$, alveolar-arterial oxygen gradient; CRF, chronic respiratory failure.

over ten years. Furthermore, our investigation revealed more crucial insights in terms of the 10-year trajectory of ABG. While ABG parameters remained stable among those who did not progress to CRF, the patients who developed CRF showed an increase in A-aDO$_2$ and a decrease in PaO$_2$ and PaCO$_2$, suggesting a phase transition before CRF progression. Retrospective tracking from the initiation of LTOT in the CRF group revealed a significant increase in A-aDO$_2$ from five years prior to LTOT initiation compared to the non-CRF group. Most notably, our data reveal a critical event occurring five years prior to LTOT initiation: a marked transition in ABG parameters, particularly A-aDO$_2$, from a stable phase to a deteriorating phase. This transition serves as a potent predictor of progression toward CRF, providing invaluable insight into the course of the disease over a decade.

Extending the follow-up of our previous prospective cohort study for a decade demonstrated the prognostic and predictive capability of A-aDO$_2$ across the entire cohort. The reason we could not sufficiently demonstrate the prognostic and predictive capability of ABG parameters within the entire cohort in our previous study might be, as indicated in this study, that COPD does not exhibit continuous deterioration but rather is thought to transition from a stable phase to a deteriorating phase at a certain point. We postulate that a six-year follow-up period may not be sufficient for patients with normal PaO$_2$ levels (PaO$_2 \geq 80$ Torr), as it may not adequately capture the period when significant deterioration occurs. Similarly, several

**Table 4. Multivariate Cox proportional hazards analysis for predicting CRF development.**

|  | CRF development HR (95% CI) | *P* value |
|---|---|---|
| A-aDO$_2$ (Torr) | 1.15 (1.07–1.24) | <0.0005 |
| ΔA-aDO$_2$ (Torr/year) | 1.15 (1.06–1.25) | <0.001 |
| K$_{CO}$ (mL/min/L/mmHg) | 0.41 (0.18–0.82) | 0.020 |
| FEV$_1$ (% predicted) | 0.98 (0.95–1.01) | 0.306 |
| Age, years | 0.98 (0.92–1.05) | 0.553 |
| BMI (kg/m$^2$) | 0.87 (0.70–1.07) | 0.174 |
| Smoking index (pack-years) | 1.01 (1.00–1.03) | 0.059 |

Multivariate model simultaneously included A-aDO$_2$, ΔA-aDO$_2$, K$_{CO}$, FEV$_1$ (% predicted), age, body mass index (BMI), and smoking index. A-aDO$_2$, alveolar-arterial oxygen gradient; BMI, body mass index; CRF, chronic respiratory failure; FEV$_1$, forced expiratory volume in 1 s; K$_{CO}$, carbon monoxide transfer coefficient.

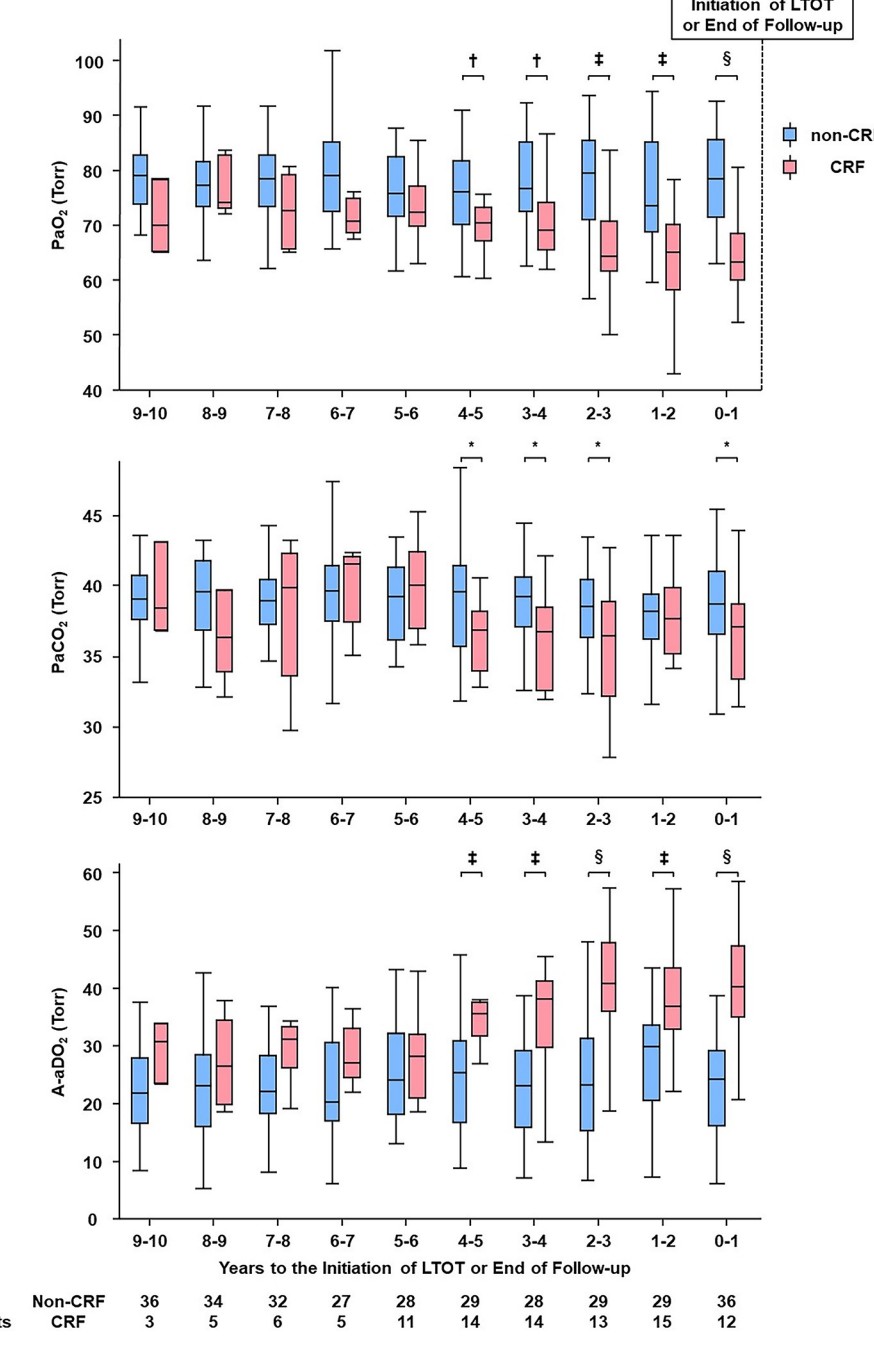

**Fig 3. Long-term changes in ABG parameters retrospectively tracked from the initiation of LTOT in the CRF group.** From five years prior to the initiation of LTOT, a decrease in $PaO_2$ and $PaCO_2$, accompanied by an increase in $A\text{-}aDO_2$, was observed. This trend significantly differed from that of the non-CRF group and was most prominent for $A\text{-}aDO_2$. $A\text{-}aDO_2$, alveolar-arterial oxygen gradient; ABG, arterial blood gas; CRF, chronic respiratory failure; LTOT, long-term oxygen therapy; $PaCO_2$, partial pressure of carbon dioxide; $PaO_2$, partial arterial pressure of oxygen. *$P<0.05$, †$P<0.005$, ‡$P<0.001$, §$P<0.0001$.

reports have indicated no temporal change in $PaO_2$ over five to six years, which is likely attributable to an insufficient follow-up period [11, 12].

COPD progression varies among individuals; even within individuals, it does not follow a uniform course. Nishimura and colleagues observed variability in the annual change in $FEV_1$ in patients with COPD over five years [13], identifying two distinct groups: 'sustainers', who maintain $FEV_1$ over a ten-year period, and 'rapid decliners', who experience a rapid reduction in $FEV_1$. Suzuki et al. also demonstrated nonuniform progression of the disease, finding that prediagnosis decline in $FEV_1$ is similar in 'rapid decliners', 'slow decliners', and 'sustainers' [14]. Furthermore, the 'sustainers' group experienced a faster initial decline in $FEV_1$ than the other groups, followed by subsequent stabilization. Besides interindividual and intraindividual heterogeneity, we observed that once deterioration begins, rapid progression to CRF requiring LTOT can occur within five years. Additional assessments revealed that there were no significant differences in the A-a$DO_2$ slope or average in the stable phase (5–10 years prior to the initiation of LTOT or the end of follow-up between the CRF and non-CRF groups), and the CRF group had significantly higher values in the deterioration phase (0–5 years prior to the initiation of LTOT) (S2 Table in S1 File). However, in the CRF group, we couldn't determine what exactly caused the transition to the deteriorating phase. Several studies indicate that acute exacerbations can trigger such a transition [15–17]. While investigating the frequency of exacerbations in the present observations, we found that in the CRF group, prevalence of yearly periods with frequent exacerbations ($\geq$2 moderate/severe exacerbations per year) were seen in 3 out of 13 (23.0%) 5–10 years prior to the initiation of LTOT and 12 out of 21 (57.1%) within the 5 years prior. In the non-CRF group, 11 out of 36 (30.6%) experienced frequent exacerbations during 5–10 years and 0–5 years before the end of follow-up. These findings suggest a possible link between acute exacerbations and the development of CRF, consequently the transition of A-a$DO_2$. However, given that a substantial number of individuals in the non-CRF group also experienced frequent exacerbations, predicting progression to CRF based solely on this is not feasible, emphasizing the importance of monitoring A-a$DO_2$.

ABG measurement is generally not recommended for routine use; it is suggested by GOLD for assessment, especially when $SpO_2$ is low [1]. However, considering that a rapid subsequent decline can occur once ABG parameters start to deteriorate, careful evaluation is necessary if any deterioration beyond a certain threshold is observed. Specifically, in the non-CRF group, the proportions of individuals with A-a$DO_2 \geq$35 Torr and $\geq$40 Torr remained at approximately 10% and 5%, respectively, even within five years before the end of follow-up. In contrast, these proportions increased in the CRF group to between 50–70% and 30–50%, respectively, within five years prior to initiation of LTOT. As there were several individuals in the non-CRF group with an A-a$DO_2 \geq$40 Torr, it is not an absolute indicator. However, we speculate that an excess of A-a$DO_2$ above these thresholds may increase the risk of developing CRF within five years. This information provides a valuable reference for determining the need for ABG follow-up. Notably, the Japanese Respiratory Society (JRS) guidelines currently do not provide specific recommendations regarding the timing of ABG measurements or the clinical significance of A-a$DO_2$ [18]. Our findings highlight the potential importance of establishing recommendations for these parameters to enhance the prediction and management of CRF progression in Japan.

Another significant insight from our research involves the natural longitudinal progression of ABG parameters in patients with COPD. In patients who developed CRF, combined downward trends of both $PaO_2$ and $PaCO_2$ contributed to the increase in A-a$DO_2$. Although a decrease in $PaO_2$ was an expected outcome, the observed declining trend constitutes a noteworthy aspect of our study. A decrease in $CO_2$ is postulated to result from hyperventilation compensating hypoxemia, and several reports have indicated an association between

hypocapnia and an increased mortality rate among patients with COPD [19–21]. As a consequence of increased minute ventilation, it can be speculated that hypocapnia leads to respiratory muscle fatigue, accelerated ventilatory failure, and ultimately death. During development of COPD, these physiological consequences may reflect a relative inability to adapt the breathing pattern to prevent respiratory muscle fatigue [19]. Despite the importance of respiratory dysfunction, such as hypercapnia, only one case required noninvasive positive pressure ventilation due to hypercapnia, and no patients developed hypercapnia before CRF. Short-term and long-term oxygen therapy has been reported to exacerbate hypercapnia [22, 23]. In this study, we did not investigate $PaCO_2$ after the initiation of LTOT, which might be related to the observed absence of significant increases in $PaCO_2$ without CRF. Moreover, we speculate that the natural course of CRF development in patients with COPD is type 1 CRF first, followed by type 2 (hypercapnic) CRF.

This study has several limitations. First, it was conducted in a single institution with a limited number of cases. We also calculated the sample size using data from our present study, and we could estimate that approximately 51 subjects per group (a total of 101 subjects) would be required. Therefore, we can conclude that we had a sufficient sample to find significances in the present study. Second, there was an increase in missing data as time progressed from the initial evaluation. Third, ABG analyses were not performed after the introduction of LTOT. However, importantly, ABG parameters during oxygen therapy differ from those evaluated in ambient air, making it difficult to assess them in the same manner. Fourth, although image analysis is significant in predicting the progression of COPD [24–27], we did not apply it in this study. This is because the primary objective of our research was to observe changes in ABG parameters, and examining their correlation with images would deviate from this main goal. Fifth, our study had missing data at various time points during the observation period, which could have affected the accuracy of the analysis. To address this, we included only completers in the non-CRF group and performed the analyses restricted to patients with at least three evaluations during both the stable phase (5–10 years prior to LTOT initiation or the end of follow-up) and the deterioration phase (0–5 years prior). The results of these analyses, detailed in S2 Table in S1 File, yielded consistent findings, supporting the robustness of our conclusions. Although a linear mixed-effects model for longitudinal data could be more suitable, our smaller sample size may have limited the feasibility of this complex analysis. Nonetheless, it is important to emphasize that our study yielded clinically significant results. The trend in A-aDO$_2$ and its potential as a prognostic factor for CRF development were clear and consistent, even with the simpler exploratory analysis used. Sixth, we did not evaluate sleep-disordered breathing or nocturnal hypoventilation. The longitudinal trend of ABG parameters demonstrated in this study may be related to sleep-disordered breathing. However, the subjects of this study had a low BMI, and thus hypoventilation due to obesity was unlikely. While this is consistent with the characteristics of patients previously reported in Japan [28, 29], the results of this study may not necessarily be generalizable to all individuals with obesity. Evaluation of sleep-disordered breathing might be necessary in such patients. Lastly, the potential overestimation of $P_AO_2$ values due to hypocapnia, which affects the accuracy of the A-aDO$_2$ calculated using the alveolar gas equation, must be considered. This is particularly relevant in COPD, where the assumption that the partial pressure of carbon dioxide in the alveoli approximates the arterial partial pressure of carbon dioxide is not valid due to ventilation/perfusion mismatch. Despite these limitations, monitoring the longitudinal changes in A-aDO$_2$ remains valuable as it minimally reflects changes in both $PaO_2$ and $PaCO_2$ levels, providing a more stable measure under varying respiratory conditions.

## Conclusions

An increasing A-aDO$_2$ trend indicates potential CRF development in patients with COPD. A transition of the annual trend of A-aDO$_2$ from a stable state to a deterioration phase can serve as a prognostic factor for CRF progression within 5 years.

## Supporting information

**S1 Checklist. STROBE statement—Checklist of items that should be included in reports of observational studies.**
(DOCX)

**S1 File.**
(DOCX)

## Acknowledgments

All authors have approved the final version of the manuscript. The authors thank Michiyoshi Nishioka, Kunihiko Terada, Tadashi Ohara, Daisuke Kinose, Megumi Naka-Kudo, Akane Haruna, Satoshi Marumo, Hirofumi Kiyokawa, Tamaki Takahashi, Koichi Hasegawa, Yoshinori Fuseya, and Kazuya Tanimura for their assistance with the data collection.

## Author Contributions

**Conceptualization:** Susumu Sato, Shigeo Muro.

**Data curation:** Kazuma Nagata, Susumu Sato, Kiyoshi Uemasu, Naoya Tanabe.

**Formal analysis:** Kazuma Nagata.

**Funding acquisition:** Susumu Sato, Shigeo Muro.

**Investigation:** Kazuma Nagata, Toyohiro Hirai.

**Project administration:** Kazuma Nagata, Susumu Sato.

**Resources:** Kiyoshi Uemasu.

**Supervision:** Susumu Sato, Kiyoshi Uemasu, Naoya Tanabe, Shigeo Muro, Toyohiro Hirai.

**Validation:** Atsuyasu Sato.

**Visualization:** Kazuma Nagata.

**Writing – original draft:** Kazuma Nagata.

**Writing – review & editing:** Susumu Sato, Naoya Tanabe, Shigeo Muro, Toyohiro Hirai.

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
