## [Decision Letter · Decision Letter 0]

27 Dec 2024

PONE-D-24-19081Trajectory of the Arterial-Alveolar Oxygen Gradient in COPD for a decadePLOS ONE

Dear Dr. Sato,

Thank you for submitting your manuscript to PLOS ONE. After careful consideration, we feel that it has merit but does not fully meet PLOS ONE’s publication criteria as it currently stands. Therefore, we invite you to submit a revised version of the manuscript that addresses the points raised during the review process.

We look forward to receiving your revised manuscript.

Kind regards,

Alonso Soto, PhD

Academic Editor

PLOS ONE

“This work was supported by JSPS KAKENHI Grant Numbers 19K08624, 16K09536, 25461156, 21590964, 16390234, JP22K11391 and JP22H04411. This research was also supported by grants from the Intractable Respiratory Diseases and Pulmonary Hypertension Research Group of the Ministry of Health, Labour and Welfare of Japan (JPMH20FC1027, JPMH23FC1031).”

“S.S. reports a grant from Nippon Boehringer Ingelheim Co. and grants from Philips-Respironics, Fukuda Denshi, Fukuda Lifetec Keiji, and ResMed that did not pertain to the submitted work. K.N. reports a honoraria for lectures at Teijin Healthcare unrelated to the submitted work. S.M. reports a grant from ROHTO Pharmaceutical unrelated to submitted work.”

Additional Editor Comments:

SUMMARY

Abbreviations should be mentioned in full the first time they are cited (COPD, LTOT)

Background:please change the phrase "CRF is a vital pathophysiology in patients with COPD...

MEASUREMENTS: How LTOT need was assessed?

TABLES use only 3 decimals for p values.

RESULTS 

Line 188 A-aDO2 refers to baseline values? Delta A-aDO2 represents the change after 1 year? Please explain it 

Reviewers' comments:

Reviewer's Responses to Questions

**Comments to the Author**

1. Is the manuscript technically sound, and do the data support the conclusions?

Reviewer #1: Yes

Reviewer #2: Yes

2. Has the statistical analysis been performed appropriately and rigorously? 

Reviewer #1: No

Reviewer #2: Yes

3. Have the authors made all data underlying the findings in their manuscript fully available?

Reviewer #1: Yes

Reviewer #2: Yes

4. Is the manuscript presented in an intelligible fashion and written in standard English?

Reviewer #1: Yes

Reviewer #2: Yes

5. Review Comments to the Author

Reviewer #1: Congratulations to the Authors:

Overall, the manuscript adheres well to the STROBE checklist. It provides a comprehensive examination of the arterial-alveolar oxygen gradient (A-aDO2) trajectory in COPD patients, with an emphasis on its potential as a prognostic marker for chronic respiratory failure. The manuscript could improve by adding details about sample size calculations, bias mitigation strategies, and a clearer acknowledgment of funding sources. However, it successfully addresses most aspects of the STROBE checklist, making it a well-structured and methodologically sound observational study. Congratulations on an important contribution to the field.

Major Observations:

1. Sample Size (STROBE Item 10): The manuscript does not provide any sample size calculation or justification for how the sample size was determined. This is critical for validating the statistical power of the study and ensuring that the sample size is sufficient to detect significant differences in outcomes. The absence of this analysis raises concerns about the reliability of the results and their generalizability.

2. Bias (STROBE Item 9): Although potential biases, such as missing data at later follow-up points, are briefly mentioned in the discussion, the methods section lacks a thorough explanation of how these biases were addressed. Specifically, the manuscript should provide more detail on how missing data were handled and what steps were taken to minimize selection bias in the patient cohort.

3. Study Design and Timeline Clarity (STROBE Items 5 and 6a): The description of the study design could benefit from greater clarity regarding the timing of follow-up assessments. While it is mentioned that arterial blood gas (ABG) analysis was performed annually, a more detailed explanation of the exact timing of assessments and how they align with patient visits, exacerbations, or disease progression would improve the understanding of the study's longitudinal design.

Reviewer #2: explain more in discussion what it `s the impact to the country where they develop the research.

the topic is interesting , an introduction apropiate .

acceptable methodology and statistics. table of results presented according to international standards

6. PLOS authors have the option to publish the peer review history of their article (what does this mean?). If published, this will include your full peer review and any attached files.

Reviewer #1: **Yes: **Johan Azañero-Haro

Reviewer #2: No

---

## [Author Response · Author response to Decision Letter 0]

2 Jan 2025

Response to the Editor

SUMMARY

Abbreviations should be mentioned in full the first time they are cited (COPD, LTOT) Thank you for your comment. 

Response:

We have revised this as suggested.

Background: please change the phrase "CRF is a vital pathophysiology in patients with COPD… 

Response:

Following your suggestion, we revised the relevant sentence as follows: "Chronic respiratory failure (CRF) is a critical complication in patients with chronic obstructive pulmonary disease (COPD) and is characterized by an increase in the arterial-alveolar oxygen gradient (A-aDO2)." 

MEASUREMENTS: How LTOT need was assessed? 

Response:

Thank you for pointing this out. The attending physician determined the need for long-term oxygen therapy (LTOT) based on the patient's clinical condition. This decision was primarily guided by the presence of progressive resting hypoxemia, defined as an arterial oxygen tension (PaO₂)≤55 Torr or a percutaneous oxygen saturation (SpO₂)≤88%. We have included this information in the revised manuscript to clarify the assessment process.

TABLES use only 3 decimals for p values. 

Response:

We have revised this as suggested.

RESULTS 

Line 188 A-aDO2 refers to baseline values? Delta A-aDO2 represents the change after 1 year? Please explain it 

Response:

Thank you for your comment. To clarify, we have revised the text to define explicitly A-aDO₂ and ΔA-aDO₂. The updated text now reads:

"In the multivariate model, which included factors believed to be predictors of CRF development in COPD patients (e.g., KCO, FEV1, age, body mass index (BMI), and smoking index), both baseline A-aDO₂ and ΔA-aDO₂ (change over the first year) were found to be significantly associated with CRF development."

Response to the Reviewer #1

Congratulations to the Authors:

Overall, the manuscript adheres well to the STROBE checklist. It provides a comprehensive examination of the arterial-alveolar oxygen gradient (A-aDO2) trajectory in COPD patients, with an emphasis on its potential as a prognostic marker for chronic respiratory failure. The manuscript could improve by adding details about sample size calculations, bias mitigation strategies, and a clearer acknowledgment of funding sources. However, it successfully addresses most aspects of the STROBE checklist, making it a well-structured and methodologically sound observational study. Congratulations on an important contribution to the field.

Major Observations:

1. Sample Size (STROBE Item 10): The manuscript does not provide any sample size calculation or justification for how the sample size was determined. This is critical for validating the statistical power of the study and ensuring that the sample size is sufficient to detect significant differences in outcomes. The absence of this analysis raises concerns about the reliability of the results and their generalizability. 

Response:

Thank you for highlighting this point. Although a formal sample size calculation was not performed before the study due to its retrospective nature, we conducted a post-hoc sample size analysis using the observed data from the present study. Using the observed means and standard deviations (CRF group: 3.76 ± 5.24 Torr/year; non-CRF group: 0.42 ± 6.65 Torr/year), we calculated the required sample size to detect a significant difference with α=0.05 and β=0.2. The analysis indicated that approximately 51 subjects per group (a total of 101 subjects) would be needed to achieve sufficient statistical power. Since our study included 157 subjects, we consider the sample size adequate to detect significant differences. We have added this explanation to the manuscript.

Bias (STROBE Item 9): Although potential biases, such as missing data at later follow-up points, are briefly mentioned in the discussion, the methods section lacks a thorough explanation of how these biases were addressed. Specifically, the manuscript should provide more detail on how missing data were handled and what steps were taken to minimize selection bias in the patient cohort. 

Response:

Thank you for your comment. We included only completers in the non-CRF group to address potential biases related to missing data. We performed the analyses restricted to patients with at least three evaluations during the stable phase (5-10 years) and the deterioration phase (0-5 years). These analyses, detailed in Table S2, showed consistent results with our primary findings, supporting the robustness of our conclusions. We have incorporated this information into the Limitation section to provide transparency about how missing data were managed.

Study Design and Timeline Clarity (STROBE Items 5 and 6a): The description of the study design could benefit from greater clarity regarding the timing of follow-up assessments. While it is mentioned that arterial blood gas (ABG) analysis was performed annually, a more detailed explanation of the exact timing of assessments and how they align with patient visits, exacerbations, or disease progression would improve the understanding of the study's longitudinal design. 

Response:

Thank you for your valuable comment. To clarify the timing of follow-up assessments, we have revised the Measurements section to provide more detailed information. Specifically, we now explain that ABG analyses were scheduled annually, with the first follow-up conducted one year after baseline. Subsequent assessments were aligned with stable phases, defined as at least four weeks after recovery from any exacerbation, to minimize variability due to acute changes.

These revisions improve the clarity of the study design and timeline.

Response to the Reviewer #2

explain more in discussion what it’s the impact to the country where they develop the research.

the topic is interesting, an introduction appropriate.

acceptable methodology and statistics. table of results presented according to international standards 

Response:

Thank you for your insightful comment. To address this, we have added a statement in the Limitation section to emphasize the relevance of our findings to Japan. Specifically, we noted that the Japanese Respiratory Society (JRS) guidelines currently lack specific recommendations regarding the timing of ABG measurements or the clinical significance of A-aDO₂. Our findings highlight the potential importance of establishing such recommendations to improve the prediction and management of CRF progression in Japan.

---

## [Editor Report · Decision Letter 1]

15 Jan 2025

Trajectory of the Arterial-Alveolar Oxygen Gradient in COPD for a decade

PONE-D-24-19081R1

Dear Dr. Sato,

We’re pleased to inform you that your manuscript has been judged scientifically suitable for publication and will be formally accepted for publication once it meets all outstanding technical requirements.

Kind regards,

Alonso Soto, PhD

Academic Editor

PLOS ONE
---

## [Editor Report · Acceptance letter]

20 Jan 2025

PONE-D-24-19081R1 

PLOS ONE

Dear Dr. Sato, 

I'm pleased to inform you that your manuscript has been deemed suitable for publication in PLOS ONE. Congratulations! Your manuscript is now being handed over to our production team.

Kind regards, 

on behalf of

Dr. Alonso Soto 

Academic Editor

PLOS ONE